# Beyond Layout Embedding: Layout Attention with Gaussian Biases for Structured Document Understanding

**Xi Zhu, Xue Han, Shuyuan Peng, Shuo Lei, Chao Deng** and **Junlan Feng***

JIUTIAN Team, China Mobile Research Institute, Beijing, China

{zhuqian,hanxueai,pengshuyuan,leishuo,dengchao,fengjunlan}@chinamobile.com

## Abstract

Effectively encoding layout information is a central problem in structured document understanding. Most existing methods rely heavily on millions of trainable parameters to learn the layout features of each word from Cartesian coordinates. However, two unresolved questions remain: (1) Is the Cartesian coordinate system the optimal choice for layout modeling? (2) Are massive learnable parameters truly necessary for layout representation? In this paper, we address these questions by proposing **L**ayout **A**ttention with **Ga**ussian **Bi**ases (**LAGaBi**): Firstly, we find that polar coordinates provide a superior choice over Cartesian coordinates as they offer a measurement of both distance and angle between word pairs, capturing relative positions more effectively. Furthermore, by feeding the distances and angles into 2-D Gaussian kernels, we model intuitive inductive layout biases, *i.e.*, the words closer within a document should receive more attention, which will act as the attention biases to revise the textual attention distribution. LAGaBi is model-agnostic and language-independent, which can be applied to a range of transformer-based models, such as the text pre-training models from the BERT series and the LayoutLM series that incorporate visual features. Experimental results on three widely used benchmarks demonstrate that, despite reducing the number of layout parameters from millions to 48, LAGaBi achieves competitive or even superior performance. Our code is available on GitHub[1].

## 1 Introduction

Structured document understanding (SDU) has gained significant research attention in the field of intelligent document processing (Park et al., 2019; Jaume et al., 2019; Han et al., 2023). It focuses on extracting layout structures and contents from scanned or digital documents, leading to enhanced performance in several downstream tasks like form comprehension and receipt understanding.

Unlike conventional text understanding (Liu et al., 2019; Vaswani et al., 2017; Kenton and Toutanova, 2019), SDU goes beyond comprehending serialized text and requires the ability to interpret documents with diverse layouts (Xu et al., 2020; Huang et al., 2022; Powalski et al., 2021; Li et al., 2021a; Wang et al., 2022a). Documents with varying layouts often contain text fields positioned in different ways. To take advantage of existing pre-trained language models, early methods (Xu et al., 2020; Li et al., 2021a,c; Appalaraju et al., 2021; Chi et al., 2020) propose to directly add 2-D position embedding to the word embedding for each word as input for Transformer. The position embedding encode the 2-D absolute coordinates $(x_0, y_0, x_1, y_1)$ of each word in the document through multi position encoding layers, where $(x_0, y_0)$ represents the upper left point and $(x_1, y_1)$ represents the lower right point of the bounding box for each word. Some researches (Powalski et al., 2021; Hong et al., 2022; Lee et al., 2022; Xu et al., 2021a; Huang et al., 2022) further proposed that absolute positions are inefficient for representing the spatial relationships between words. They employ relative positions between words to encode spatial relationships. For example, Hong et al.(2022) map the relative positions into embeddings, which are then multiplied with the semantic embedding of the word to calculate inter-word layout scores. This score is incorporated into the self-attention layers to combine semantic and layout features.

Despite the significant progress made, we argue that current methods, whether based on absolute or relative positioning, heavily rely on a large number of trainable parameters for position embeddings from Cartesian coordinates, often comprising millions of parameters. This raises two unexplored questions: (1) Is the Cartesian coordinate system the optimal choice for layout modeling? (2) Are

---

*Junlan Feng is the corresponding author.

[1] https://github.com/zxilucky/LAGaBi

massive learnable layout parameters truly necessary? Concerning the former, various coordinate systems exist, including Cartesian, polar, and spherical coordinates, yet previous research has solely focused on Cartesian coordinates. Regarding the latter, it is intuitively expected that words closer within a document should receive more attention. However, this simple inductive bias may not be effectively learned solely through gradient updates.

In this paper, we present a unified investigation of the two aforementioned problems. Regarding the choice of coordinate systems, we find that polar coordinates offer a more efficient representation than Cartesian coordinates for expressing relative positions. By computing the differences in distance and angle between two words in polar space, polar coordinates outperform their Cartesian counterparts by providing extra angle information. For layout learning, we discover that layout modeling can be achieved by a specific distribution: words closer in space receive higher layout scores, eliminating the need for extra position embeddings. Combining these two choices, we propose LAGaBi (Layout Attention with Gaussian Biases). LAGaBi formulates pairwise spatial relationships between tokens using the distance and angle in the polar coordinate system. Moreover, the distance and angle are fed into a 2-D Gaussian distribution to output a layout score. We choose the Gaussian distribution because it guarantees that the layout score decreases when either the distance or angle variables increase, making it the most commonly used distribution for this purpose. The layout score is then incorporated into the original self-attention as attention bias, resulting in a revised distribution that considers both text and layout features. We introduce trainable Gaussian kernels to better align the semantic and layout scores at different scales, adding just $4\times$ attention heads extra parameters. For instance, based on RoBERTa (Liu et al., 2019) with 12 attention heads, there are only 48 additional parameters that need to be learned for encoding layout features.

Extensive experiments demonstrate that LAGaBi achieves remarkable performance on diverse SDU benchmarks, including FUNSD (Jaume et al., 2019), CORD (Park et al., 2019), and XFUND (Xu et al., 2021b), across both monolingual and multilingual scenarios. LAGaBi emerges as a versatile module that seamlessly integrates with transformer-based language models, such as BERT (Kenton and Toutanova, 2019), RoBERTa, and InfoXLM (Chi et al., 2020), empowering them to effectively process structured documents and achieve significant performance gains of up to 27.01 points. Additionally, LAGaBi can be incorporated into complex SDU models that leverage visual features, such as LayoutLM (Xu et al., 2020), LayoutLMv2 (Xu et al., 2021a), and LayoutLMv3 (Huang et al., 2022), leading to further performance enhancements and establishing new state-of-the-art results.

## 2 Related Works

Significant progress has recently been made by using the Transformer-based pre-trained model (PTM) to learn the cross-modality interaction between textual and layout information, which has been demonstrated to be critical for structured document understanding (Xu et al., 2020; Huang et al., 2022; Powalski et al., 2021; Li et al., 2021a; Wang et al., 2022a). LayoutLM (Xu et al., 2020) modified the input of BERT (Kenton and Toutanova, 2019) by adding position embedding layers to encode word-level 2-D coordinates, while StructualLM (Li et al., 2021a) proposed to encode segment-level positions. LiLT (Wang et al., 2022a) encoded text and layout using two different transformer layers separately and adopted bi-directional attention to fuse them. Besides, there are a series of works that use multi-modal transformers to model text, layout, and image simultaneously, such as SelfDoc (Li et al., 2021b), DocFormer (Appalaraju et al., 2021), StrucTexT (Li et al., 2021c), ERNIE-Layout (Peng et al., 2022), mmLayout (Wang et al., 2022b).

Most of the above approaches encode the 2-D absolute positions, ignoring the critical relative spatial relationships between words that are essential to textual semantic understanding. Hong et al.(2022) proposed to encode the spatial relationships as relative position embeddings, which are then multiplied with the semantic embedding of the token to calculate inter-word layout scores. This score is incorporated into the self-attention to combine semantic and layout features. GeoLayoutLM (Luo et al., 2023) introduces geometric relations and brand-new geometric pre-training tasks in different levels for learning the geometric layout representation, whose geometric relations are largely dependent on some pre-defined manual rules. TITL (Powalski et al., 2021) encodes the relative positions between words in a simpler manner, and it adopts linear layers to convert the 2-D discrete distance (implemented through bucketing)

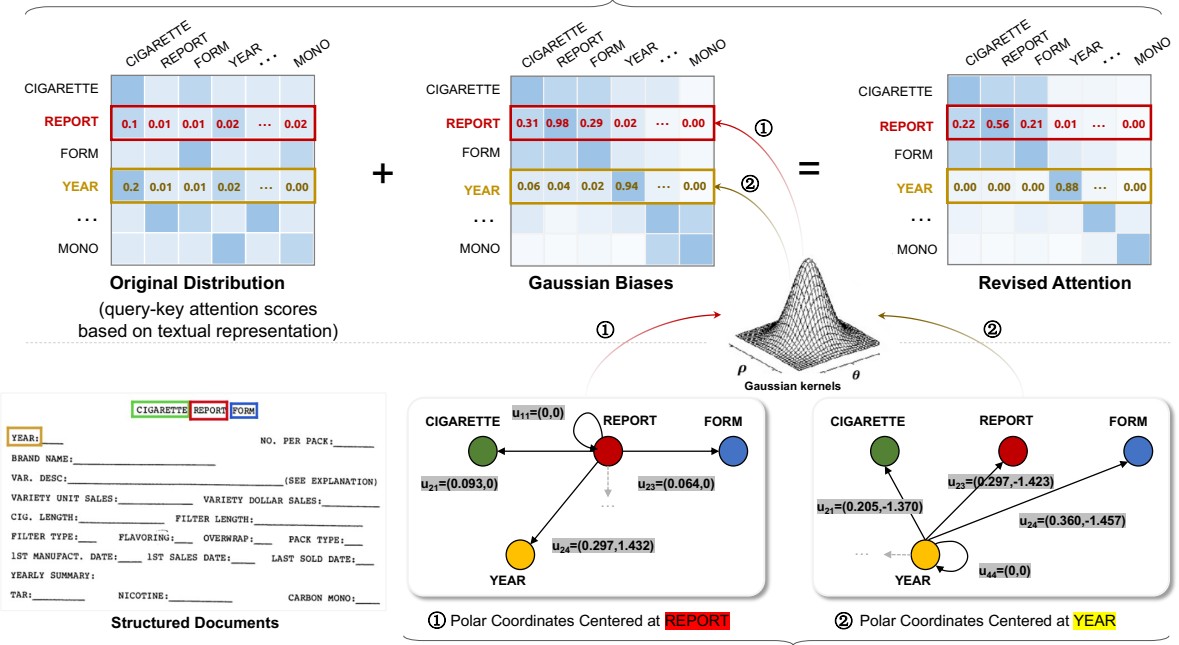

Figure 1: An illustration of our Layout Attention with Gaussian Biases (LAGaBi). For each query word, we first compute the spatial relationships of each keyword in a polar system centered on it. Then Gaussian kernels are employed to transform these spatial relationships into attention biases, which will revise the original query-key attention distribution. Such biases are shared across different Transformer layers. When using LAGaBi, we can encode the document layout structure without adding position embedding layers at the bottom of the network.

between words into attention biases motivated by T5 (Raffel et al., 2020). Such relative attention biases have also been employed by LayoutLMv2 (Xu et al., 2021a) and LayoutLMv3 (Huang et al., 2022), yielding notable results. Different from them, we model the relative positions from a new perspective, *i.e.* introducing polar coordinates with distance and angle that are more efficient in representing relative spatial relationships. Furthermore, we employ the extremely lightweight Gaussian kernels to encode polar coordinates into attention biases, offering a more streamlined approach with fewer trainable parameters and aligning better with human intuition compared to linear layers.

## 3   Methodology

Structured documents typically contain both textual words and layouts, where textual words denote the main content of the document, while layouts record the organizational form of text. Optical character recognition (OCR) as a classical technique for image document parsing can recognize the text as well as its locations. Formally, given a document $\mathcal{D}$, OCR identifies the text words

$\mathcal{W} = \{w_i\}_{i=1}^{N}$ and their associated layout positions $\mathcal{C} = \{c_i\}_{i=1}^{N}$, where $N$ is the number of words while $c_i = [x_i^0, y_i^0, x_i^1, y_i^1]$ represents the left-top and right-bottom coordinates of the bounding box that contains the $i_{th}$ word. An ideal model for document understanding should take both text words and their layout positions into consideration.

The key to structured document comprehension, building upon the achievements of text understanding techniques such as Transformer (Vaswani et al., 2017), lies in effectively representing document layouts while remaining compatible with language representation. Therefore, there are two crucial problems that need to be solved: 1. How to model document layouts efficiently? 2. How to integrate the layouts into Transformer to guide textual semantic understanding of the document content?

In this paper, we introduce a novel layout modeling method for structured document understanding, namely Layout Attention with Gaussian Biases (LAGaBi). As shown in Figure 1, it mainly consists of two parts: modeling relative positions with polar coordinates and layout attention with Gaussian biases. The former is responsible for modeling the 2-D relative positions between words by po-

lar coordinates which are capable of representing inter-word spatial relationships; The latter focuses on transforming polar coordinates into attention biases, which will modify the original semantic query-key attentions between words into a more suitable distribution that accurately captures the underlying layout structure of documents.

### 3.1 Spatial Relationships with Polar Coordinates

In a structured document, the relative spatial relationships between words are shown to be more important than their absolute coordinates, which can assist humans in better understanding pair-wise semantic dependencies. For example, tokens in the same line generally have stronger semantic associations with each other, while tokens that are farther away and on different lines are more difficult to form strong associations. Although some previous works (Powalski et al., 2021; Xu et al., 2021a; Li et al., 2021a) have proposed to model the relative distances, which acted as learnable attention biases, they still rely on learnable positional embeddings and only utilize the relative horizontal and vertical distances in Cartesian coordinate system.

Different from previous works, we propose to capture inter-word spatial relationships through polar coordinates, in which both orientations and distances can be preserved. For each query token, we can build a polar coordinate system centered at its position, and calculate the polar coordinates (spatial relationships) of its keys. More concretely, the position of the query token is regarded as the reference point (pole), and the horizontal direction in the current Cartesian coordinate system is set as the reference direction (polar axis) following the most common reading habit, *i.e.*, left-to-right and top-to-down. Formally, given a query token with its 2-D coordinates $c_i$ as the pole, the polar coordinates $\mathbf{u}_{ij} = (\rho_{ij}, \theta_{ij})$ of the $j_{th}$ key in this document page can be calculated as below:

$$\rho_{ij} = \sqrt{(x_j - x_i)^2 + (y_j - y_i)^2} \qquad (1)$$

$$\theta_{ij} = tan^{-1}((y_j - y_i)/(x_j - x_i)) \qquad (2)$$

where $\rho_{ij} \in [0, 1]$ and $\theta_{ij} \in [-\pi/2, \pi/2]$ denote the distance and angle (orientation) from the $i_{th}$ token to the $j_{th}$ token respectively, and $(x_i, y_i)$ are the normalized coordinates of the top-left point of the $i_{th}$ bounding box. For instance, in Figure 1, when taking "REPORT" as the reference point, the spatial relationship between the words "FORM" and "REPORT" can be represented as a polar coordinate (0.064, 0), indicating a distance of 0.064 and an angle of 0 degrees, while that between the words "YEAR" and "REPORT" is (0.297, 1.432).

### 3.2 Layout Attention with Gaussian Biases

How to use the essential relative spatial relationships to guide the model to perceive layout information is a problem worth exploring. Inspired by ALiBi (Press et al., 2021) and T5 bias (Raffel et al., 2020) that encode the 1-D relative position information as attention biases upon the query-key scores instead of positional embedding, we propose to revise attention scores/distribution with 2-D attention biases that integrate spatial relationships. Specifically, the attention score in a single-head self-attention can be modified as:

$$a_{ij} = \frac{\exp(\mathbf{q}_i \mathbf{k}_j^T / \sqrt{d_k} + \alpha\,(\mathbf{g}(\mathbf{u}_{ij}) - 1))}{\sum_{j=1}^{N} \exp(\mathbf{q}_i \mathbf{k}_j^T / \sqrt{d_k} + \alpha\,(\mathbf{g}(\mathbf{u}_{ij}) - 1))} \qquad (3)$$

where $\mathbf{q}_i$ is the $i_{th}$ query vetor, $\mathbf{q}_i$ is the $j_{th}$ key vetor, and $d_k$ is the dimension of the attention head. $\mathbf{g}(\mathbf{u}_{ij})$ denotes the attention biases, which is derived from the 2-D Gaussian kernel with learnable parameters based on polar coordinates $\mathbf{u}$ indicating spatial relationships. The Gaussian kernel ensures that words farther within the document are assigned a smaller layout score. Therefore, by incorporating a reversed term $(\mathbf{g}(\mathbf{u}_{ij}) - 1)$, we can significantly penalize the attention scores of query-key pairs that are farther, while making only slight revisions to the scores of closer pairs. $\alpha$ is a hyper-parameter that makes a trade-off between semantic association and spatial dependency. It denotes how much the spatial relationship between the key and the query contributes to their semantic association. In particular, a convenient formula of $\mathbf{g}(\mathbf{u})$ is:

$$\mathbf{g}(\mathbf{u}) = exp(-\frac{1}{2}(\mathbf{u} - \boldsymbol{\mu})^T \boldsymbol{\Sigma}^{-1}(\mathbf{u} - \boldsymbol{\mu})) \qquad (4)$$

where $\boldsymbol{\Sigma}$ and $\boldsymbol{\mu}$ are learnable 2 × 2 and 2 × 1 covariance matrix and mean vector of a Gaussian kernel, respectively. We further restrict the covariances to have diagonal form, resulting in 2 × 2 parameters per kernel for each attention head. Note that the Gaussian kernels are different across different attention heads, but are shared across different self-attention layers. Thus, there is a total of 2 × 2 × $N_{heads}$ learnable parameters for our attention biases, where $N_{heads}$ denotes the attention head

number in each self-attention layer. For example, taking the RoBERTa base as the backbone, there are 48 parameters that need to be learned. Notably, we only include layout information in the keys and queries but not in the values, ensuring that the text semantics are not corrupted.

**Properties of our LAGaBi:** (1) It is easy to implement and can be adapted to any transformer-based model without changing its structure. (2) The position embedding layer is discarded, and there are very few parameters to learn, which can be done during the fine-tuning stage, so it is quite efficient. (3) It decouples layout and text understanding, allowing the potential of language models to be fully exploited in structured document understanding.

## 4 Experiments

### 4.1 Datasets

**Pre-training Dataset.** Following LayoutLM (Xu et al., 2020), we also pre-train our model using the IIT-CDIP Test Collection 1.0 (Lewis et al., 2006), which is a large-scale dataset with over 11 million scanned document images. Only 1 million of them are used for fast pre-training. We pre-process each document page using Tesseract[2], an open-source OCR engine, to retrieve the textual contents as well as their layouts. We normalize the coordinates of each token to integers in the range of 0 to 1000 and add an empty bounding box $[0, 0, 0, 0]$ to the special tokens [CLS], [SEP], and [PAD].

**Fine-tuning Datasets.** We evaluate our method on both monolingual (English) and multilingual document information extraction datasets listed below. **FUNSD** (Jaume et al., 2019) is a form dataset that uses forms to extract and organize textual information. It contains 199 documents, 149 of which are for training and 50 of which are for testing. **CORD** (Park et al., 2019) is a dataset for receipt key information extraction that includes 800, 100, and 100 receipts for training, validating, and testing, respectively. **XFUND** (Xu et al., 2021b) is a multilingual version of FUNSD with 8 languages, each language containing 199 instances (149 for training and 50 for testing) as FUNSD.

### 4.2 Implemention Details

Our approach is model-agnostic and language-independent, which can be applied to a range of transformer-based models. In this paper, we have evaluated our method based on three kinds

[2]https://github.com/tesseract-ocr/tesseract

of baselines: 1) monolingual models (BERT (Kenton and Toutanova, 2019) and RoBERTa (Liu et al., 2019)), 2) multilingual model (InfoXLM (Chi et al., 2020)), and 3) document understanding models (LayoutLM (Xu et al., 2020), LayoutLMv2 (Xu et al., 2021a), and LayoutLMv3 (Huang et al., 2022)). Gaussian kernels will be included in the self-attention blocks of each model, which describe the relative spatial relationship between tokens.

**Pre-training.** We only conduct pre-training tasks for the two monolingual models: BERT+LAGaBi and RoBERTa+LAGaBi. We initialize the weight of them with the corresponding baselines, except the Gaussian kernels. The parameters of the Gaussian kernels, namely covariance matrixes and mean vectors, are randomly initialized. Both models are simply supervised by masked language modeling (MLM) loss during pre-training. Adam optimizer (Kingma and Ba, 2014) is adopted with a learning rate of $5e-5$, weight decay of $1e-2$ and $(\beta_1, \beta_2) = (0.9, 0.999)$. The batch size is set to 128 and all the two models are trained for 200,000 steps on 8 NVIDIA v100 32GB GPUs.

**Fine-tuning.** In this paper, we mainly focus on the document understanding task of semantic entity labeling, which aims at assigning each semantic entity a BIO label. We add a token-level classification layer upon the base models (including monolingual, multilingual, and document understanding models) to predict the BIO labels for this task. Word-level F1 score is adopted as the evaluation metric. The fine-tuning process takes 2000 steps using a batch size of 16 and the Adam optimizer with a learning rate of 5e-5 for FUNSD and 7e-5 for CORD and XFUND. Fine-tuning configurations for document understanding models follow their official releases. Hyper-parameter $\alpha$ is set to 4 for all experiments, which has been tuned on the CORD's val set.

### 4.3 Experimental Results

#### 4.3.1 Performance on monolingual datasets

We first evaluate our method on the monolingual form and receipt understanding datasets. From the results shown in Table 1, we can observe that:
(1) **The lightweight LAGaBi enables simple integration with language models, allowing them to effectively process structured documents**. For example, without any pre-training on document data, RoBERTa+LAGaBi has achieved 84.84% and 95.97% F1 scores on FUNSD and CORD, surpassing baseline model RoBERTa by 18.38% and

| Model | #Parameters | Pre-training | Modality | FUNSD | CORD |
|---|---|---|---|---|---|
| BERT (Kenton and Toutanova, 2019) | 110M | - | T | 60.26 | 89.68 |
| RoBERTa (Liu et al., 2019) | 125M | - | T | 66.48 | 93.54 |
| BROS (Hong et al., 2022) | 110M | 11M | T+L | 81.21 | - |
| FormNet (Lee et al., 2022) | 217M | 11M | T+L | 84.69 | - |
| LiLT (Wang et al., 2022a) | 131M | 11M | T+L | 88.41 | 96.07 |
| **BERT+LAGaBi**(w/o pre-train) | 110M+48 | - | T+L | 74.14 (+13.88) | 93.44 (+3.76) |
| **BERT+LAGaBi**(w/ pre-train) | 110M+48 | 1M | T+L | 87.27 (**+27.01**) | 95.82 (**+6.14**) |
| **RoBERTa+LAGaBi**(w/o pre-train) | 125M+48 | - | T+L | 84.84 (+18.36) | 95.97 (+2.43) |
| **RoBERTa+LAGaBi**(w/ pre-train) | 125M+48 | 1M | T+L | **89.15** (+22.67) | **96.56** (+3.02) |
| LayoutLM (Xu et al., 2020) | 160M | 11M | T+L+I | 79.27 | 94.31[*] |
| LayoutLMv2 (Xu et al., 2021a) | 200M | 11M | T+L+I | 82.70 | 94.95 |
| StrucTexT (Li et al., 2021c) | 107M | 11M | T+L+I | 83.09 | - |
| DocFormer (Appalaraju et al., 2021) | 183M | 11M | T+L+I | 83.34 | 96.33 |
| LayoutLMv3 (Huang et al., 2022) | 133M | 11M | T+L+I | 90.29 | 96.56 |
| **LayoutLM+LAGaBi**(w/o pre-train) | 160M+48 | - | T+L+I | 87.77 (+8.10) | 94.93(+0.62) |
| **LayoutLMv2+LAGaBi**(w/o pre-train) | 200M+48 | - | T+L+I | 88.16 (+5.49) | 97.05(+2.10) |
| **LayoutLMv3+LAGaBi**(w/o pre-train) | 133M+48 | - | T+L+I | **91.00** (+0.71) | **97.05** (+0.49) |

Table 1: Performance on FUNSD and CORD for monolingual structured document understanding. "T/L/I" denotes the "text/layout/image" modality. (+x) denotes the gain in F1 score compared to base model, while [*] show the result from our re-implementation. All the F1 scores in percentage (%) are reported.

2.43% respectively. RoBERTa+LAGaBi also outperforms several representative document understanding models such as LayoutLM (Xu et al., 2020) and LayoutLMv2 (Xu et al., 2021a). The results show that LAGaBi is a powerful method for capturing essential layout features, allowing language models to be easily extended to adapt to structured document understanding tasks.

(2) **Pre-training brings profits**. After pre-training on 1 million unlabeled document data, our method exhibits extra improvements. RoBERTa+LAGaBi with pre-train surpasses all other approaches except LayoutLMv3 (Huang et al., 2022). While there is still a minor difference on FUNSD between our model and LayoutLMv3, our method is significantly easier to implement, introducing only 48 extra learnable parameters to the vanilla Transformers structure, making it more computationally efficient and flexible.

(3) **The LAGaBi could also be seamlessly coupled with other layout embedding and multi-modal-based document understanding models, improving their performance even further.** Performance improvements on LayoutLM (Xu et al., 2020), LayoutLMv2 (Xu et al., 2021a) and LayoutLmv3 (Huang et al., 2022)) are obvious, with F1 score gains of 8.10%, 5.49%, 0.71% on FUNSD and 0.62%, 2.10%, 0.49% on CORD. The process

is particularly efficient since it only requires fine-tuning based on the published pre-trained weights rather than pre-training from scratch. Furthermore, by combining LAGaBi with the top-performing LayoutLMv3 model, we achieve new state-of-the-art results on both FUNSD and CORD datasets.

### 4.3.2 Performance on multilingual dataset

Following the multi-lingual LayoutXLM (Xu et al., 2021b) and LiLT (Wang et al., 2022a), we also evaluate our method based on InfoXLM (Chi et al., 2020) on three sub-tasks: language-specific fine-tuning, multi-task fine-tuning, and zero-shot transfer learning. We first perform fine-tuning based on the Gaussian kernels with random initialization. For a fair comparison, following LiLT, we also adopt the pre-trained Gaussian kernels for further fine-tuning, and we employ the Gaussian kernels in RoBERTa+LAGaBi which have been pre-trained on 1M monolingual document data in Sec 4.3.1. Results on XFUND are shown in Table 2.

**LAGaBi is also valid in multilingual scenarios, allowing multilingual language models to understand structured documents and outperform existing best-performing methods**. LAGaBi can largely increase the performance of the multilingual language model InfoXLM on all three tasks, regardless of whether the Gaussian kernels are

| Task | Model | Pre-train Data | FUNSD | XFUND | | | | | | | Avg. |
|------|-------|----------------|-------|-------|----|----|----|----|----|----|------|
| | | size & language | EN | ZH | JA | ES | FR | IT | DE | PT | |
| **Language-specific** | XLM-RoBERTa (2020) | - | 66.70 | 87.74 | 77.61 | 61.05 | 67.43 | 66.87 | 68.14 | 68.18 | 70.47 |
| | InfoXLM (2020) | - | 68.52 | 88.68 | 78.65 | 62.30 | 70.15 | 67.51 | 70.63 | 70.08 | 72.07 |
| | LayoutXLM (2021b) | 30M-Mutli | 79.40 | 89.24 | 79.21 | 75.50 | 79.02 | 80.82 | 82.22 | 79.03 | 80.56 |
| | LiLT (2022a) | 11M-Mono | 84.15 | 89.38 | 79.64 | **79.11** | 79.53 | 83.76 | 82.31 | 82.20 | 82.51 |
| | **InfoXLM+LAGaBi** | - | 83.35 | 89.38 | 84.01 | 78.06 | 82.80 | 84.40 | **84.95** | 83.25 | 83.78 |
| | **InfoXLM+LAGaBi** | **1M-Mono** | **84.17** | 89.65 | 84.73 | 77.51 | 83.86 | 85.19 | 83.89 | 83.47 | 84.06 |
| **Zero-shot** | XLM-RoBERTa (2020) | - | 66.70 | 41.44 | 30.23 | 30.55 | 37.10 | 27.67 | 32.86 | 39.36 | 38.24 |
| | InfoXLM (2020) | - | 68.52 | 44.08 | 36.03 | 31.02 | 40.21 | 28.80 | 35.87 | 45.02 | 41.19 |
| | LayoutXLM (2021b) | 30M-Mutli | 79.40 | 60.19 | 47.15 | 45.65 | 57.57 | 48.46 | 52.52 | 53.90 | 55.61 |
| | LiLT (2022a) | 11M-Mono | 84.15 | **61.52** | 51.84 | 51.01 | 59.23 | 53.71 | 60.13 | 63.25 | 60.61 |
| | **InfoXLM+LAGaBi** | - | 83.35 | 47.91 | 46.13 | 48.98 | 54.84 | 47.40 | 53.23 | 58.82 | 55.08 |
| | **InfoXLM+LAGaBi** | 1M-Mono | **84.17** | 39.88 | 35.77 | 44.82 | 54.45 | 46.72 | 51.64 | 56.37 | 51.73 |
| **Multi-task** | XLM-RoBERTa (2020) | - | 66.33 | 88.30 | 77.86 | 62.23 | 70.35 | 68.14 | 71.46 | 67.26 | 71.49 |
| | InfoXLM (2020) | - | 65.38 | 87.41 | 78.55 | 59.79 | 70.57 | 68.26 | 70.55 | 67.96 | 71.06 |
| | LayoutXLM (2021b) | 30M-Mutli | 79.24 | 89.73 | 79.64 | 77.98 | 81.73 | 82.10 | 83.22 | 82.41 | 82.01 |
| | LiLT (2022a) | **1M-Mono** | 85.74 | 90.47 | 80.88 | **83.40** | 85.77 | 87.92 | 87.69 | 84.93 | 85.85 |
| | **InfoXLM+LAGaBi** | - | **86.67** | 90.90 | 86.49 | 81.74 | 86.44 | 87.74 | **88.18** | 86.72 | 86.89 |
| | **InfoXLM+LAGaBi** | **1M-Mono** | 86.35 | **92.00** | 86.86 | 81.50 | **87.00** | 87.96 | 87.58 | 87.24 | 87.22 |

Table 2: The performance on FUNSD and XFUND with different settings, including language-specific fine-tuning (fine-tuning on X, testing on X), zero-shot transfer (fine-tuning on FUNSD, testing on X), and multi-task fine-tuning (fine-tuning on all 8 languages, testing on X). "1M-Mono" denotes 1 million monolingual (English) documents used for pre-training, while "30M-Multi" is the multilingual version. All the $F_1$ scores in percentage (%) are reported.

pre-trained or not. InfoXLM+LAGaBi using the pre-trained Gaussian kernels outperforms the top-performing method LiLT on both language-specific and multi-task fine-tuning tasks, with average F1 scores of 84.06% and 87.22%, demonstrating the efficacy of LAGaBi in multilingual scenarios. On the zero-shot transfer learning task, LAGaBi fell slightly behind its counterparts. This may be due to the inherent gaps between different languages, such as differences in reading order and semantic density. For example, English usually uses spaces to separate words and has uneven word lengths, while Chinese appears as a tighter sequence with smaller semantic units (*i.e.*, characters). Such layout knowledge learned from a specific language shows limited contributions to other languages. This phenomenon also proves the effectiveness of LAGaBi in modeling layouts, *i.e.* it actually has acquired the layout knowledge for a specific language after fine-tuning on the corresponding data.

## 4.4 Ablation Studies

To investigate the impacts of our learnable Gaussian kernels and polar coordinates, we have con-ducted extensive ablation experiments based on several RoBERTa variants equipped with different layout encoding mechanisms (*e.g.* layout embedding layers, linear bias layers, and fixed/learnable Gaussian Kernels), and spatial relationships (*e.g.* distance, angle, and 2D-xy distance). All the variants are evaluated without further pre-training, and only fine-tuned for 2000 steps on FUNSD and CORD. Results on FUNSD's test set and CORD's validation set are shown in Table 3. From the result, we can observe that LAGaBi with learnable Gaussian kernels and polar coordinates (#8) can significantly outperform the baseline (#1) and the model with linear layout embedding layers (#2), indicating that models encoding layouts as attention biases are superior to layout embedding-based methods.

**Impact of the Gaussian kernels.** To study the effects of various methods for converting polar coordinates to attention biases, we compared linear layers (#3), fixed Gaussian kernels (#4), and learnable Gaussian kernels (#8). The results demonstrate that linear layers are far less effective than Gaussian kernels, which is likely due to the fact that the Gaussian kernels are more in line with human

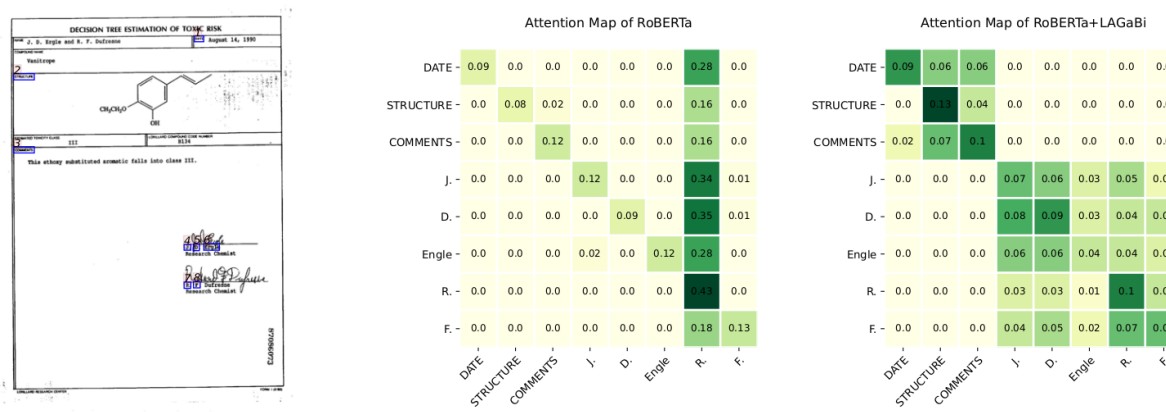

Figure 2: Visualization of the attention maps. The word-word attention scores are obtained by aggregating token-level attention in the last layer of transformers. We also annotate the position and order of each word in the original document page, and the attention scores are rounded to two decimal places for better visualization.

| Task | # | Ablation Strategy | FUNSD | CORD |
|---|---|---|---|---|
| | 1 | RoBERTa(baseline) | 66.48 | 92.29 |
| | 2 | + embedding layers | 69.59 | 92.67 |
| Impact of Gaussian | 3 | + linear bias layers | 76.32 | 93.00 |
| | 4 | + fixed kernels | 83.81 | 93.00 |
| Impact of Polar-Coor. | 5 | + Euclidean distance | 73.05 | 93.72 |
| | 6 | + Angle | 84.48 | 94.70 |
| | 7 | + 2D-xy distance | 79.85 | 94.21 |
| | 8 | + LAGaBi | **84.84** | **94.77** |

Table 3: Ablation studies on the effectiveness of the learnable Gaussian kernels and the polar coordinates.

intuition than linear layers. Learnable Gaussian kernels (#8) also achieve better performance than fixed Gaussian kernels whose mean is 0 and variance is 1 (#4), since the learnable Gaussian kernels enjoy better flexibility to adapt to different formats. **Impact of the polar coordinates.** Polar coordinates, which consist of two elements: distance and angle, is a typical technique for describing spatial relationships. We analyzed the effects of distance (#5) and angle (#6), as well as the classical 2-D relative horizontal and vertical distances proposed by TITL (Powalski et al., 2021) (#7). The results suggest that the model with angle information (#6) is more effective than the model with distance information (#5 and #7). We hypothesize that this is because angles are less impacted by size scaling than distances, but they are more sensitive to location changes. Furthermore, LAGaBi achieves much better performance than the model that employs horizontal and vertical distances, which further reveals the superiority of polar coordinates.

## 4.5 Analysis

**Impact of $\alpha$.** Hyper-parameter $\alpha$ makes a trade-off between semantic and layout contributions when computing pair-wise attention scores in our LABaBi, which is important. We conduct several experiments with different $\alpha$ settings to study the impact of $\alpha$ based on RoBERTa without any further pre-training. $F_1$ scores on CORD's validation set are listed in Table 4, showing the model achieves its best performance when $\alpha = 4$.

| $\alpha =$ | 0 | 1 | 2 | 3 | 4 | 5 |
|---|---|---|---|---|---|---|
| **CORD** | 92.29 | 92.82 | 93.08 | 93.46 | **94.77** | 94.39 |

Table 4: Impact of different $\alpha$ settings. All the scores are from CORD's validation set.

**Visualization analysis.** We visualize the word-level attention maps of the baseline RoBERTa and our RoBERTa+LAGaBi. Due to space limitations, in this paper, we only show the attention maps of the first 8 words in the input sequence. The case in Figure 2 is from the test set of FUNSD. As shown in Figure 2, the RoBERTa incorrectly associates the character "R." with all the words, while most other words are treated as unrelated. According to the attention map in the RoBERTa+LAGaBi, greater attention scores arise between "R." and "F.", "J." and "D.", all of which are placed in the signature area, whereas attention scores between remote irrelevant words such as "DATE" and "STRU" are zeros. This demonstrates that LAGaBi indeed learns more accurate semantic associations by incorporating layout information. More examples and detailed analysis can be seen in Appendix.A.

# 5 Conclusion

In this paper, we propose a model-agnostic and language-independent method that leverages Layout Attention with Gaussian Biases to encode the relative spatial positions for structured document understanding (SDU). Specifically, we first model the inter-word spatial relationships using polar coordinates. Then the query-key attention scores are revised by the Gaussian biases that are related to their spatial relationships. Our method can be applied to a series of Transformer-based models with extremely few parameters, improving their performance for SDU tasks. Experiments based on six transformer-based SDU models and three monolingual/multilingual benchmarks fully demonstrate the effectiveness of our proposal. This research provides new ideas for structured document understanding tasks, which are expected to promote the efficient development of document intelligence.

## Limitations

Despite the superior performance exhibited by LAGaBi, it does have some limitations. Firstly, in our experiments with the LayoutLM series that integrate multi-modal features, LAGaBi was only fine-tuned for validation without pre-training. We believe that leveraging multi-modal pre-training could further improve LAGaBi's performance based on LayoutLM, and this will be explored in future investigations. Secondly, although we have empirically demonstrated the effectiveness of polar coordinates and Gaussian distribution in layout learning, our motivation is driven by a simple intuition rather than rigorous mathematical proof.

## Acknowledgements

We sincerely thank all the anonymous reviewers for their valuable comments and suggestions.

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

# A Appendix

Figure 3 in this section shows the attention distribution of four additional examples on FUNSD. From all the examples, it can be observed that the RoBERTa model typically treats each word in isolation, while RoBERTa+LAGaBi can learn the more accurate correlations between different words based on their layout relationships.

We also visualize the word-level attention maps of LayoutLMv3 and LayoutLMv3+LAGaBi. LayoutLMv3 is currently the top-performing method for structured document understanding, which utilizes the 2-D relative positions through linear attention biases. From the attention maps shown in Figure 4, we can observe that LayoutLMv3+LAGaBi refers more to layout information when modeling the inter-word semantic correlations, while LayoutLMv3 is relatively independent. For example, in the third sample of Figure 4, LAGaBi learns more dense associations among "Tiers", "II.", and "&" in the neighborhood than LayoutLMv3, which is in line with human intuition.

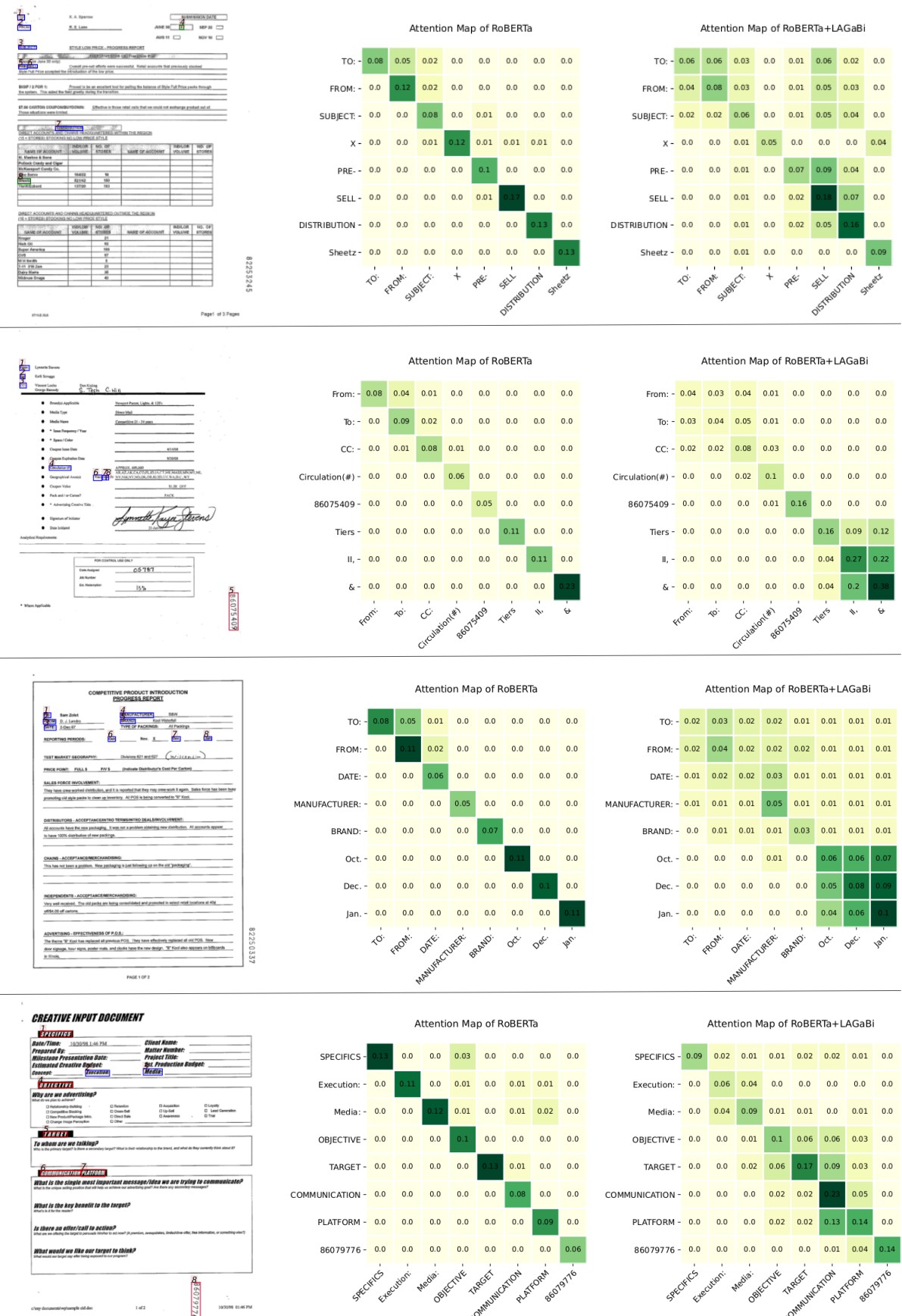

Figure 3: Visualization of more examples based on RoBERTa.

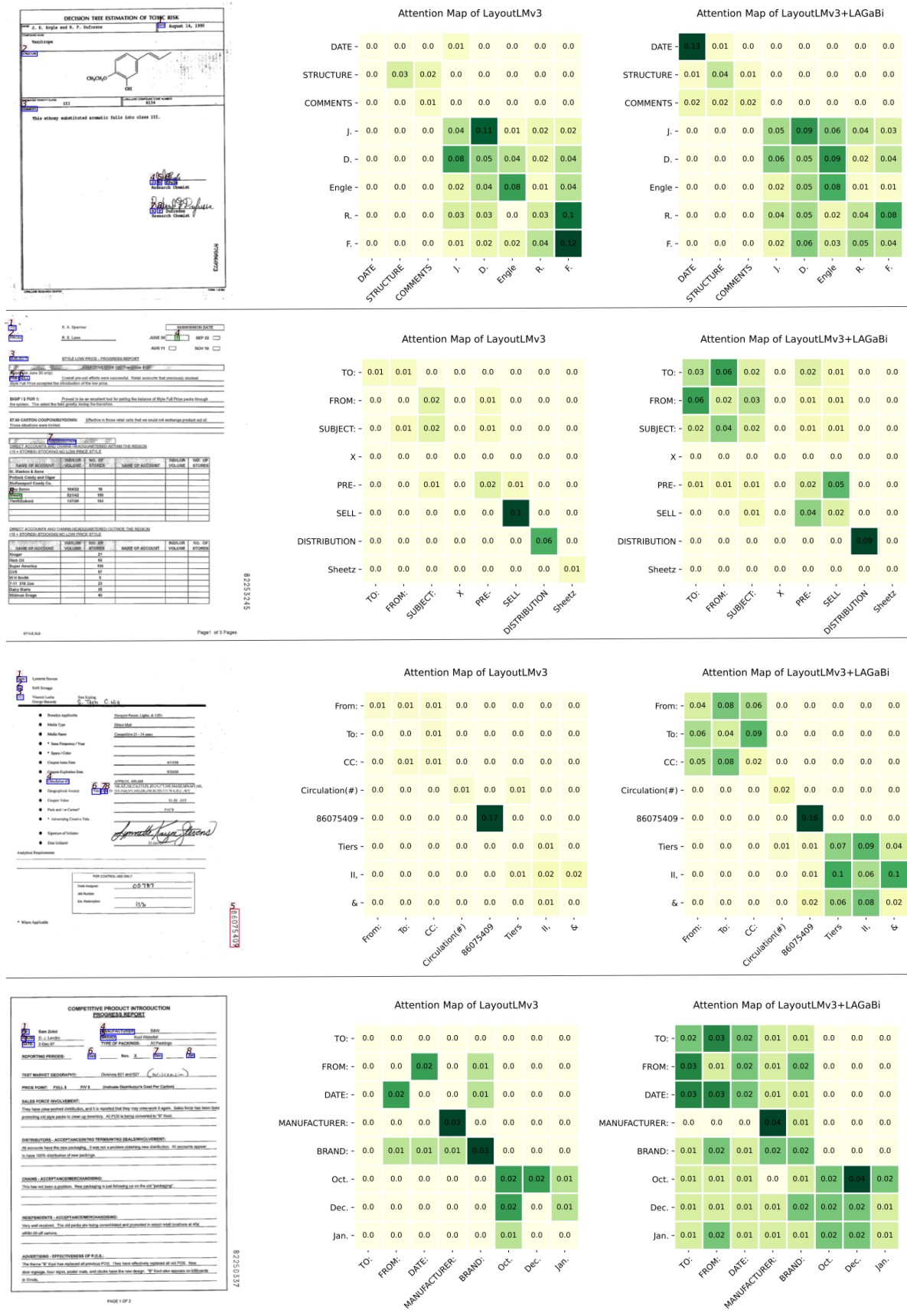

Figure 4: Visualization of examples base on LayoutLMv3.