# OpenReview forum: "Beyond Layout Embedding: Layout Attention with Gaussian Biases for Structured Document Understanding"
_EMNLP/2023/Conference — EMNLP 2023 Findings_

### Official Review · Reviewer_NEpE · 2023-08-04

**Soundness:** 4

**Excitement:**

3: Ambivalent: It has merits (e.g., it reports state-of-the-art results, the idea is nice), but there are key weaknesses (e.g., it describes incremental work), and it can significantly benefit from another round of revision. However, I won't object to accepting it if my co-reviewers champion it.

**Paper Topic And Main Contributions:**

This paper focuses on addressing the problem of efficient document layout representation learning for structured document understanding. It conducts an investigation into the efficiency of current methods, such as the Cartesian coordinate system and potential redundant training parameters. Based on the observations and analyses, the authors propose a new approach called "Layout Attention with Gaussian Biases" for the task of structured document understanding. The proposed method aims to enhance efficiency and effectiveness in capturing document layouts, and provides a promising insights for improving the representation learning process in this context.

**Reasons To Accept:**

A new method is proposed to address the structured document understanding (SDU) task, along with an investigation into two unresolved questions in this domain.

The proposed method can be flexibly combined with existing PLMs.

The proposed method brings performance gains for PLMs on SDU tasks.

**Reasons To Reject:**

Marginal performance improvement for some backbone models (e.g., LayoutLMv3 in Table1, and the performance in several languages in Table2.).

Some critical comparison are missing. (e.g., LayoutLM v1-v3 W/ pre-training).

**Reproducibility:**

3: Could reproduce the results with some difficulty. The settings of parameters are underspecified or subjectively determined; the training/evaluation data are not widely available.

**Reviewer Confidence:**

4: Quite sure. I tried to check the important points carefully. It's unlikely, though conceivable, that I missed something that should affect my ratings.

---

> ### Author Rebuttal · Authors · 2023-08-29
>
> We thank for the reviewer's positive evaluation such as the flexible method, promising insights, effective results, etc. There are also very valuable questions we try to answer as follows.
>
> >Marginal performance improvement for some backbone models (e.g., LayoutLMv3 in Table1, and the performance in several languages in Table2.).
>
> Our method achieves state-of-the-art results on all tasks, including FUNSD, CORD, and XFUND. In response to the reviewer's concerns, we provide the following explanations:
>
> LayoutLMv3 is currently the top-performing model for document understanding, which has achieved performance close to the upper bound of FUNSD and CORD. By incorporating our LAGaBi, the F1 scores on Funsd and CORD are further improved from 90.29% and 96.56% to 91.00% and 97.05% in Table 1, respectively, achieving the state-of-the-art with negligible extra computational cost.
>
> In the multilingual scenario, InfoXLM serves as the backbone. InfoXLM+LAGaBi brings significant improvements on all three tasks: language-specific fine-tuning, zero-shot transfer, and multi-task fine-tuning, with average F1 gains of **11.99%**, **10.54%**, and **16.16%**, which is a highly remarkable achievement. We have excerpted the results of InfoXLM and InfoXLM+LAGaBi (w/ pre-training) from Table 2:
> | Task | Model |EN |  ZH |  JA |  ES |  FR | IT | DE | PT | Avg. |
> | :-----| :----- | :----: | :----: |  :----- | :----: | :----: |  :----- | :----: | :----: | :----: |
> | language-specific fine-tuning | InfoXLM | 68.52 |  88.68 | 78.65 |  62.30 | 70.15 | 67.51 | 70.63 | 70.08 | 72.07 |
> || InfoXLM+LAGaBi |  84.17|  89.65 | 84.73 | 77.51 | 83.86 | 85.19 | 83.89 | 83.47 | 84.06|
> || **gains**| **+15.65** | **+0.97** | **+6.08** |**+15.21** |**+13.71** |**+17.68** |**+13.26**|**+13.39** |**+11.99**
> ||
> | zero-shot transfer | InfoXLM |68.52 |  44.08 |  36.03 |  31.02 |  40.21  | 28.80 |  35.87 |  45.02 |  41.19 |
> || InfoXLM+LAGaBi |   84.17  | 39.88 |  35.77  | 44.82  | 54.45  | 46.72 |  51.64 |  56.37  | 51.73|
> || **gains**| **+15.65** | -4.2 | -0.26 |**+13.8** |**+14.24** |**+17.92** | **+15.77** |**+11.35** | **+10.54**|
> ||
> | multi-task fine-tuning | InfoXLM | 65.38 |  87.41 |  78.55  | 59.79 |  70.57  | 68.26  | 70.55 |  67.96  | 71.06 |
> || InfoXLM+LAGaBi |  86.35 |  92.00 |  86.86 |  81.50 |  87.00 |  87.96 |  87.58  | 87.24 |  87.22|
> || **gains**| **+20.97** | **+4.59** | **+8.31** |**+21.71**|**+16.43** |**+19.70** | **+17.03** |**+19.28** | **+16.16**|
>
> According above results, LAGaBi has shown significant effectiveness across all tasks and languages, except for Chinese and Japanese in the zero-shot transfer scenario. Zero-shot transfer in this paper refers to fine-tuning on an English dataset (FUNSD) and testing on the target language. English and Chinese/Japanese exhibit distinct differences in reading order and semantic dependencies. For instance, English typically uses spaces to separate words and has uneven word lengths, while Chinese have tighter sequencing with smaller semantic units (i.e., characters). Such variations in layout and semantic structures between English and Chinese/Japanese limit the effectiveness of learned layout knowledge from English when applied to Chinese/Japanese. This phenomenon also validates the efficacy of LAGaBi in modeling layouts, as it efficiently acquires language-specific layout knowledge after fine-tuning on corresponding datasets.
> We have discussed this point in Sec.4.3.2 (from lines 479-492) of our summited paper.
>
> >Some critical comparison are missing. (e.g., LayoutLM v1-v3 W/ pre-training).
>
> Thank you for appreciating and having high expectations for our work.
>
> The core contribution of this paper is the proposal of the new paradigm in layout modeling, LAGaBi, which decouples the process of modeling layouts and texts, enabling transformer-based language models to understand structured data in a straightforward manner. This simple and pluggable layout modeling approach allows the task of structured document understanding to be independent of specific model architectures, languages, and scale variations.
>
> Through our research, we have showcased significant improvements on 3 public monolingual/multilingual document datasets for 3 transformer-based language models (i.e. BERT, Roberta, and InfoXLM) with/without pretraining. Additionally, we have evaluated our method based on 3 powerful multi-modal transformer-based models (i.e. LayoutLMv1~v3) by fine-tuning, impressive performance on 2 datasets also demonstrating LAGaBi's ability. These results can strongly support the our claims and contributions.
>
> Due to limitations in computational resources (only 8 V100 GPUs), we were unable to conduct pre-training on the multi-modal Transformer models. We have mentioned this issue in the Limitations section. We are looking forward to acquiring enough resources for pre-training multi-modal models in the future, and we also encourage more researchers to validate and extend our method, contributing to the advancement of the entire community.

---

### Official Review · Reviewer_zPEC · 2023-08-04

**Soundness:** 3

**Excitement:**

3: Ambivalent: It has merits (e.g., it reports state-of-the-art results, the idea is nice), but there are key weaknesses (e.g., it describes incremental work), and it can significantly benefit from another round of revision. However, I won't object to accepting it if my co-reviewers champion it.

**Missing References:**

1. ERNIE-Layout: Layout Knowledge Enhanced Pre-training for Visually-rich Document Understanding(https://arxiv.org/pdf/2210.06155.pdf)
2. StructuralLM: Structural Pre-training for Form Understanding(https://arxiv.org/pdf/2105.11210.pdf)

**Paper Topic And Main Contributions:**

This paper focuses on the task of structured document understanding and proposes a method that utilizes layout information. Unlike previous document pre-training methods that involve overlaying layout embeddings onto input embeddings, this approach incorporates the relative positional relationships between layout elements directly into the attention matrix using Gaussian function. The approach demonstrates positive outcomes in both transformers and multi-modal transformers. Specifically, the method transforms the representation of layout information from Cartesian coordinates to polar coordinates to compute the relative positional relationships between bounding boxes. Subsequently, it models the layout bias using Gaussian function and directly adds the obtained results to the attention matrix.

**Questions For The Authors:**

Question A：Why are there no experiments conducted on the DocVQA dataset?
Question B：What are the differences between the layout bias in LAGaBi and the Spatial-Aware Self-Attention Mechanism in LayoutLMv2?
Question C: Why are the scores for distant word pairs all 0 in the attention matrix, which is inconsistent with the actual situation?

**Reasons To Accept:**

- The layout information is represented in polar coordinates and transformed into layout bias using Gaussian function, which directly influence the attention score without requiring additional layout embedding. This approach significantly reduces the number of parameters compared to the layout embedding method.
- The proposed approach has demonstrated significant improvements on FUNSD and CORD datasets for both transformer and multi-modal transformer models.

**Reasons To Reject:**

- The innovation is somewhat limited. Using polar coordinates and Gaussian function to calculate the relative position relationship matrix for bounding boxes is essentially analogous to computing 2D relative positions in document pre-training models like LayoutLMv2.
- The evaluation datasets used in this paper are FUNSD and CORD, both of which are sequence labeling tasks. However, no comparison was made with metrics on the document-based question answering dataset, like DocVQA.
- The experiment is incomplete. Why is there no pre-training experiment on the multi-modal transformer?

**Reproducibility:**

4: Could mostly reproduce the results, but there may be some variation because of sample variance or minor variations in their interpretation of the protocol or method.

**Reviewer Confidence:**

4: Quite sure. I tried to check the important points carefully. It's unlikely, though conceivable, that I missed something that should affect my ratings.

---

> ### Author Rebuttal · Authors · 2023-08-29
>
> We thank for the reviewer's positive evaluation such as the lightweight method, significant improvement, etc, There are also very valuable questions and advice we try to answer as follows.
> >The innovation is somewhat limited. Using polar coordinates and the Gaussian function to calculate the relative position relationship matrix for bounding boxes is essentially analogous to computing 2D relative positions in document pre-training models like LayoutLMv2.
>
> This is a very insightful opinion, and we try to make more detailed explanations as follows :
>
> Indeed, both LAGaBi and the Spatial-Aware Self-Attention Mechanism (SASAM) in LayoutLMv2 utilize relative positions, which has been a hot topic in NLP recently. However, it is important to note that there are *fundamental differences between them in terms of **motivations**, **methodologies**, and **impacts***.
>
> **1.Different motivations.**
>
> Our main motivation is to decouple position encoding from word embeddings and eliminate the need for overlay layout embedding layers. This allows the task of structured document understanding to be independent of specific model architectures, languages, and scale variations. On the other hand, SASAM aims to incorporate relative positional information on top of position embeddings, taking inspiration from the relative position encoding strategy in T5.
>
> **2.Different methodologies.**
>
> *(a)Different methods for representing relative positions.* LAGaBi using relative polar coordinates to model the relative positions between different words from a new perspective, modeling distances as well as directions. It aligns well with human reading habits and proves to be more sensitive to layout changes compared to solely considering distances. Conversely, SASAM employs a heuristic strategy for modeling relative positions by using discrete distances in the vertical and horizontal directions (implemented through bucketing), which is reminiscent of the 1-D relative positions commonly used in NLP tasks.
>
> *(b)Different strategies for mapping relative positions.* LAGaBi adopts simple Gaussian kernels to transform the relative positions (polar coordinates) into attention biases, with just 4× attention heads extra parameters to be learned. Moreover, this approach aligns better with human intuition, as it assigns larger correlation weights (attention biases) to word pairs with smaller distances or angles within a document page. SASAM, on the other hand, uses linear layers to convert relative positions into attention biases, requiring more parameters and a larger amount of training data to learn a proper mapping function that aligns well with human reading habits.
>
> **3.Different impacts.**
>
> LAGaBi, based on RoBERTa without any visual features, achieves impressive F1 scores of 89.15% and 96.56% on the FUNSD and CORD datasets, respectively, surpassing the multi-modal LayoutLMv2 with SASAM by 6.45% and 1.61%. Furthermore, LAGaBi can be effectively combined with SASAM to further enhance the model's performance. For instance, by incorporating LAGaBi, the F1 scores of LayoutLMv2 on the FUNSD and CORD datasets can be improved by 5.49% and 2.10%, respectively.
>
> Overall, the novelty lies in the important scenario where the technique is introduced, the effectiveness it enables, as well as potential changes and insights it inspires. We will add these discussions in our revised version to make this point more clear.
>
> >The evaluation datasets used in this paper are FUNSD and CORD, both of which are sequence labeling tasks. However, no comparison was made with metrics on the document-based question-answering dataset, like DocVQA.
>
> Thanks for such a constructive suggestion.
> We primarily focus on the text-centric tasks in structured documents, specifically information extraction tasks, and we have achieved significant results on both monolingual and multilingual scenarios. Additionally, we have also conducted experiments on tasks such as document question answering (DocVQA) and document image classification (RVL-CDIP) based on Roberta, with the following results:
>
> | Method | DocVQA (ANLS) | RVL-CDIP(accuray) |
> | :----- | :----: | :----: |
> | Roberta| 66.42 | 90.06 |
> | **Roberta+LAGaBi** (w/o pre-training) | **70.31** | **90.59**|
>
> The results show that our method also brings improvements on DocVQA and RVL-CDIP over baseline (Roberta) without pre-training,  further supporting the claim that LAGaBi can make language models able to understand structured documents more accurately.
>
> >Why is there no pre-training experiment on the multi-modal transformer?
>
> Thank you for appreciating and having high expectations for our work.
>
> The core contribution of this paper is the proposal of the new paradigm in layout modeling, LAGaBi, which decouples the process of modeling layouts and texts, enabling transformer-based language models to understand structured data in a straightforward manner. This simple and pluggable layout modeling approach allows the task of structured document understanding to be independent of specific model architectures, languages, and scale variations.
>
> Through our research, we have showcased significant improvements on 3 public monolingual/multilingual document datasets for 3 transformer-based language models (i.e. BERT, Roberta, and InfoXLM) with/without pretraining. Additionally, we have evaluated our method based on 3 powerful multi-modal transformer-based models (i.e. LayoutLMv1~v3) by fine-tuning, impressive performance on 2 datasets also demonstrating LAGaBi's ability. These results can strongly support the our claims and contributions.
>
> Due to limitations in computational resources (only 8 V100 GPUs), we were unable to conduct pre-training on the multi-modal Transformer models. We have mentioned this issue in the Limitations section. We are looking forward to acquiring enough resources for pre-training multi-modal models in the future, and we also encourage more researchers to validate and extend our method, contributing to the advancement of the entire community.
>
> > Why are the scores for distant word pairs all 0 in the attention matrix, which is inconsistent with the actual situation?
>
> This is due to the rounding operation for better visualization, where the attention scores are rounded to two decimal places. In this case, attention scores for distant word pairs less than 0.01 are truncated to 0. As we can see in Figure 2, word pairs with an attention score displayed as 0 indeed have weak or no strong semantic correlations between them. Therefore, the rounding operation does not affect the understanding of the model's visualization results. To avoid any potential confusion, we will make a note of this in the final version.

---

### Official Review · Reviewer_MynQ · 2023-08-10

**Soundness:** 4

**Excitement:**

3: Ambivalent: It has merits (e.g., it reports state-of-the-art results, the idea is nice), but there are key weaknesses (e.g., it describes incremental work), and it can significantly benefit from another round of revision. However, I won't object to accepting it if my co-reviewers champion it.

**Missing References:**

[1] mmLayout: Multi-grained MultiModal Transformer for Document Understanding. ACM MM 2022.

[2] ERNIE-Layout: Layout Knowledge Enhanced Pre-training for Visually-rich Document Understanding. Findings of EMNLP 2022.

[3] GeoLayoutLM: Geometric Pre-training for Visual Information Extraction. CVPR 2023.

**Paper Topic And Main Contributions:**

The authors propose the Layout Attention with Gaussian Biases. They leverage the polar coordinates to capture the layout information in the document images. Furthermore, they model an intuitive inductive layout bias, the words closer within a document should receive more attention, by feeding the distances and angles into a 2-D Gaussian kernel. They conduct experiments on three document image information extraction datasets to evaluate the performance of their method.

**Questions For The Authors:**

A: What are the impacts of Gaussian kernel and polar coordinates on the CORD?

B: Compared with methods based on cartesian coordinates, what are the advantages of polar coordinates?

Please refer to the comments in the Reasons To Reject for details.

**Reasons To Accept:**

1. The idea of replacing the Cartesian coordinates with the polar coordinates is interesting.
2. The paper is well written and easy to understand.
3. The main experiment results show the proposed method is effective.

**Reasons To Reject:**

1. Ablation experiments were only conducted on one dataset. It would be better to conduct ablation experiments on more datasets. For example, what are the impacts of Gaussian kernel and polar coordinates on the CORD?
2. Compared with methods based on cartesian coordinates, what are the advantages of polar coordinates? I hope that the authors can provide a more in-depth analysis. For example, the authors compare the attention maps of RoBERTa and RpBERTa+LAGaBi. However, I think comparing the attention maps of LayoutLMv3 and LayoutLMv3+LAGaBi would be more meaningful for understanding the effect of LAGaBi.
3. The authors miss some related work (see Missing References).

**Reproducibility:**

2: Would be hard pressed to reproduce the results. The contribution depends on data that are simply not available outside the author's institution or consortium; not enough details are provided.

**Reviewer Confidence:**

5: Positive that my evaluation is correct. I read the paper very carefully and I am very familiar with related work.

---

> ### Author Rebuttal · Authors · 2023-08-29
>
> We thank for the reviewer's positive evaluation such as interesting ideas, effective results, etc. There are also very valuable questions and advice we try to answer as follows.
> >Ablation experiments were only conducted on one dataset, It would be better to conduct ablation experiments on more datasets. For example, what are the impacts of Gaussian kernel and polar coordinates on the CORD?
>
> Thanks for such a constructive suggestion. We have conducted ablation experiments on the CORD dataset following the settings depicted in Sec. 4.4, and the experimental results on the validation set are shown in the table below.
> | Task | # | Ablation Strategy |  F1 score on CORD |
> | :-----| :---- | :---- | :----: |
> |   | 1 | RoBERTa(baseline) | 92.29 |
> |   | 2 |  + layout embedding layers | 92.67 |
> |  Impact of Gaussian | 3 |  + linear bias layers | 93.00 |
> |   | 4 |  + fixed kernel  | 93.95 |
> |  Impact of Polar-Coor. | 5 |  + Euclidean distance | 93.72 |
> |   | 6 |  + Angle | 94.70 |
> |   | 7 |  + 2D-xy distance |  94.21 |
> |  **ours** | **8** |  **+ LAGaBi** | **94.77** |
>
> The ablation results on the CORD dataset align with the findings on the FUNSD dataset (as illustrated in Table 3 in Sec. 4.4). This suggests that both the Gaussian kernel and polar coordinates play a role in enhancing the baseline model's ability to comprehend structured documents. Thanks for your advice, and we will include these ablation results in Table 3 in our final version.
>
> >Compared with methods based on cartesian coordinates, what are the advantages of polar coordinates? I hope that the authors can provide a more in-depth analysis.
>
> This is a very thought-provoking question, and we try to explain it as follows：
>
> Polar coordinates offer a representation of spatial relationships in terms of distance and angle, with the angle component being particularly valuable in capturing the relative direction between words within structured documents. This representation provides an intuitive understanding of how words are positioned with respect to each other. Considering common writing and reading habits (such as left-to-right and top-to-down), words within the same line tend to exhibit stronger semantic connections, while words on different lines are typically less semantically related. When taking a specific word as an anchor, words that are located farther away from the reference reading direction (e.g., words on different lines) generally exhibit larger relative angles compared to words along the reading direction. Thus, polar coordinates, which include angle information, prove to be more sensitive to layout changes compared to Cartesian coordinates that solely rely on distances.
> Furthermore, based on the findings from the ablation studies (as demonstrated in Table 3 in Section 4.4), it is evident that Polar coordinates surpass Cartesian coordinates in effectively modeling structured documents.
> >However, I think comparing the attention maps of LayoutLMv3 and LayoutLMv3+LAGaBi would be more meaningful for understanding the effect of LAGaBi.
>
> Comparing the attention maps based on LayoutLMv3 is indeed a promising idea. We will incorporate this analysis in the appendix of the final version.
>
> >The authors miss some related work (see Missing References)
>
> Thanks for the detailed advice! We have also noticed these excellent works you mentioned, and we will include discussions of these related works in our revised version.
>
> >About reproducibility
>
> Our method only depends on the simple learnable Gaussian kernels and does not involve introducing complex model structures, training objectives, or additional datasets. All experiments were conducted on publicly available datasets, making them easily reproducible. Once the paper is accepted, we will promptly release our source code and models on Github or HuggingFace repository, to foster the development of the entire community.

---

### Meta-Review · Area_Chair_KUJf · 2023-09-18

**Recommendation:** 4

**Metareview:**

The authors introduce a novel attention module that leverages polar coordinates instead of cartesian coordinates to capture the layout of the images in the documents. While the reviewers raised some concerns about the experimental setup, most of them seem to have been addressed in the rebuttal.

---

### Decision · Program_Chairs · 2023-10-07

**Decision:**

Accept-Findings

**Comment:**

The authors introduce a novel attention module that leverages polar coordinates instead of cartesian coordinates to capture the layout of the images in the documents. While the reviewers raised some concerns about the experimental setup, most of them seem to have been addressed in the rebuttal.